# Evaluating citizen science outreach: A case-study with The Peregrine Fund's American Kestrel Partnership

Sarah E. Schulwitz[1][◐]*, Greg C. Hill[2][‡], Vanessa Fry[3][‡], Christopher J. W. McClure[1][◐]

1 The Peregrine Fund, Boise, Idaho, United States of America, 2 School of Public Service, Boise State University, Boise, Idaho, United States of America, 3 Idaho Policy Institute, Boise State University, Boise, Idaho, United States of America

◐ These authors contributed equally to this work.
‡ These authors also contributed equally to this work.
* Schulwitz.Sarah@peregrinefund.org

**Data Availability Statement:** All relevant data are within the manuscript and its Supporting information files.

## Abstract

Citizen science programs can be powerful drivers of knowledge and scientific understanding and, in recent decades, they have become increasingly popular. Conducting successful research with the aid of citizen scientists often rests on the efficacy of a program's outreach strategies. Program evaluation is increasingly recognized as a critical practice for citizen science practitioners to ensure that all efforts, including outreach, contribute to the overall goals of the program. The Peregrine Fund's American Kestrel Partnership (AKP) is one such citizen science program that relies on outreach to engage participants in effective monitoring of a declining falcon species. Here, we examine whether various communication strategies were associated with desired outreach goals of the AKP. We demonstrate how social media, webcams, discussion boards, and newsletters were associated with perception of learning, agreement with our conservation messaging, and participation in our box monitoring program. Our results thus help us to improve our outreach methodology, suggest areas where other citizen science programs might improve their outreach efforts, and highlight future research priorities.

## Introduction

Citizen science, or public participation in research [1], can be a powerful driver of knowledge and scientific understanding [2–4]. Citizen science programs are generally multi-faceted endeavors that must simultaneously focus on research and outreach [1]. Importantly, successful research using citizen scientists often rests on the efficacy of a program's outreach strategies.

Outreach efforts of citizen science programs must achieve several objectives including participant recruitment, training, and retention. Many programs rely on recruitment of participants with a minimum level of knowledge of the subject matter who are willing to learn and follow a standardized protocol [5]. Training regarding field techniques and data collection

**Funding:** SES received an award from the McClure Family Foundation (https://www.mcclurefamilyfoundation.org/) specific to funding this work. The funders had no role in study design, data collection and analysis, decision to publish, or preparation of the manuscript.

**Competing interests:** The authors have declared that no competing interests exist.

protocol is often accomplished via websites, videos, and material posted online. For example, Galaxy Zoo (https://www.zooniverse.org/projects/zookeeper/galaxy-zoo/) provides a tutorial and field guide to help participants classify galaxies in photographs. Retention of participants is also important because quality of data collected by volunteers likely increases as they gain experience [6, 7]. Citizen scientists are often motivated by a desire to learn about their focal subjects [8–10]. Indeed, retention of these volunteer scientists can be enhanced by the prospect of continued learning [11].

Social media is thought to be an important avenue for scientific collaboration and communication [12] and can aid in recruitment of citizen scientists while fostering an online community [13, 14]. Similarly, social media, online blogs, and chat forums can also aid in participant learning [15, 16]. Social media and online discussion platforms can therefore be valuable tools for practitioners of citizen science.

Webcams are common, yet understudied, outreach and educational tools [17]. There are hundreds of webcams available for viewing on the internet, many of which are dedicated to STEM (science, technology, engineering, and mathematics) education [18]. Yet, most published work on STEM webcams focuses on them as a data collection tool [17, 19–22]. Given the ubiquity of webcams, research into their efficacy in accomplishing goals of citizen science programs is a priority.

Phillips et al. 2014 [23] argue that program evaluation should be of great importance for citizen science practitioners to ensure that all efforts, including outreach, contribute to the overall goals of the program. Regrettably, there is little research from citizen science programs on how participants prefer to communicate [10] or which communication methods are most effective. Thus, case studies evaluating and reporting on lessons learned in the practice of citizen science outreach are badly needed to generate a body of knowledge that can guide future work.

The Peregrine Fund is a research based conservation organization with a mission to conserve raptors, or birds of prey. Historically, the organization worked with highly endangered species with only a few individuals. Management and data acquisition relied on a few focused staff members or teams at a few sites, for example with the California Condor (*Gymnogyps californianus*) or island endemic Ridgway's hawk (*Buteo ridgwayi*). As we expanded to include more widespread but still jeopardized species in our work, such as American Kestrels (*Falco sparverius*), we recognized the need for a new data acquisition strategy. For species with continent-wide distributions, like the American Kestrel, we recognized that data collection must also be widespread. We determined that engaging citizen scientists could serve as a promising mechanism to get the data we sought. Thus, in 2012 we launched the American Kestrel Partnership (AKP) as The Peregrine Fund's first citizen science program.

Early years of the program found us, mostly biologists by training, exploring this domain that was new to our organization. We sought to educate, recruit and train our partners who in turn, we hoped, would provide high quality data on American Kestrels on a continental scale. Thus, we implemented a plethora of engagement opportunities that we hoped would help us meet our objectives, including several social media accounts, the KestrelCam, a discussion board, and a newsletter.

The Peregrine Fund launched the AKP with the ultimate goal of elucidating drivers of declines in populations of American Kestrels (*Falco sparverius*) across much of North America [24, 25]. Collecting data at scales sufficient to examine continental population declines would be cost-prohibitive without the aid of citizen scientists. AKP thus coordinates the installation and monitoring of kestrel nest boxes by citizen scientists. Outreach goals therefore include recruiting participants, informing them of kestrel biology and current understanding of population trends, training them to follow protocols, retaining their participation, and soliciting financial donations. To accomplish these goals, we disseminated outreach material regarding

kestrel biology, instructions for nest box assembly, detailed monitoring protocols, and data collection sheets via the AKP website (kestrel.peregrinefund.org) and electronic newsletters. We further implemented an online discussion forum, a live-streamed nest cam (hereafter, "KestrelCam") [17, 19, 20] and social media accounts with the goal of distributing outreach material, directing the public to the website, and recruiting citizen scientists.

Here, we examine whether various communication strategies as implemented by the AKP are associated with our desired outreach goals. Using an online survey, we assessed (1) demographics of our partners and followers, (2) how our partners and followers interact with our program's components (box monitoring program, KestrelCam, social media, newsletter, discussion board), (3) how interaction with program components is related to interaction with other program components (e.g., do KestrelCam watchers also monitor boxes?), and (4) how interacting with program components is related to perception of learning or knowledge on the species' plight. The results of this case study will inform our own program operation but should also be useful to other practitioners interested in program design and strategic evaluation.

## Materials and methods

The study was exempt from IRB review due to it being research that included a survey in which responses were obtained in a manner in which the identity of the human subjects was not able to be ascertained directly or through identifiers linked to the respondents. In addition, any disclosure of the responses would not reasonably place the respondents at risk.

### Target audience and survey design and dissemination

At the time of survey creation, we were in regular contact with 3,710 constituents that were receiving newsletter updates about AKP. Constituents had either registered as a partner on the AKP website (n = 1230), signed up through The Peregrine Fund's website to receive updates on the AKP or the KestrelCam (n = 2330), or both (n = 150). We sent a survey to all of these constituents.

AKP partnered with Boise State University's Idaho Policy Institute (IPI) to design, disseminate and analyze the survey. We designed the survey using the online platform, Qualtrics, and emailed surveys directly to constituents. The survey was distributed in English only and was designed to be 10 minutes or less in duration because survey abandonment rates increase after roughly eight minutes [26]. A skip-logic, branching design automatically enabled respondents to only see and answer questions pertinent to their experience. In other words, a certain response on one question would bring the respondent to the next relevant question, allowing them to skip irrelevant questions. We tested the survey with team members plus several non-project staff members to test that our wording was clear and easy to understand and to ensure our skip-logic flowed as intended. The best method for increasing likelihood of participation is a pre-email by a known party describing the intent of the survey [27]. We thus emailed a notification to the 3,710 constituents described above two days prior to survey dissemination describing the relationship between IPI and AKP, the need for the survey, and requesting the recipients' participation in the survey. The survey was in the field for four weeks from June 2-June 30, 2017. A mid-point survey reminder email to recipients who had not yet completed the survey.

### Survey content

Our survey assessed various aspects of constituent participation, including whether respondents monitor or own a kestrel nest box, have registered with the AKP, enter their nest and

observation data through the website (if applicable), follow AKP protocols (if applicable), or watch the KestrelCam. Respondents were asked to indicate the level of importance of each of our communication methods: newsletters, KestrelCam, social media (Facebook and Twitter), and discussion board. For communication preference questions, respondents selected five for modes that were very important and one for least important. Respondents were also asked if they would be willing to use a mobile app for data entry, and about basic demographics (age, education, gender).

We assessed respondents' perceptions of having learned about kestrel conservation through the AKP's efforts with the following question: "Do you feel that you have learned about American Kestrel biology and/or efforts by The Peregrine Fund to understand the decline of the American Kestrel?" Response options were "yes" or "no." We also assessed respondents' alignment with our primary conservation science message. Regarding this question: we have found that the ability for artificial nest sites to benefit a population is conditional on site quality and population demographics [28, 29]. For example, artificial cavities of sufficient quality and placement would benefit a nest site-limited population [29]. Alternatively, "bad boxes" (i.e., poor quality or poorly placed boxes) increase mortality or reduce reproductive success [e.g., 28]. The introduction of too many "bad boxes" could cause a stable or increasing population to decline [29]. Through outreach, we explained this concept. We further explained that proper monitoring enables partners to recognize a "bad box." Once recognized, the partner can remove or relocate a "bad box" to reduce potential harm. Our primary conservation science message, therefore, was "Installing nest boxes is not necessarily a golden ticket to helping kestrels: not all boxes help, and some boxes could harm a population. That is why we encourage partners that have a box to commit to monitoring and submitting data." We communicated this in the two quarterly newsletters prior to survey dissemination. We assessed alignment with our primary message with the question: "Do you think installing nest boxes will reverse the kestrel decline? Response options were "yes," "no," 'I don't know,' and "other."

See S1 Appendix for full survey.

## Data interpretation and analyses

To gauge participation in the box-monitoring program, we scored a series of questions related to participating in the nest box program from which we calculated a "participation score" for each respondent. Participation scores were calculated by giving a point for each positive response to the five following questions. "Do you currently have a kestrel box?," "Are you registered as an AKP partner?," "Have you registered your box(es) with AKP's nest box database?," "Do you input your monitoring data into the AKP database?," and "Do you follow AKP's recommendations for monitoring boxes when checking your box(es)?" Possible participation scores ranged from zero to five.

To determine the overall level of importance for communication types, we calculated the percentage of each score for each communication type and calculated a single average score for each communication type. To determine if differences existed between the overall ranks, we performed Kruskal-Wallis test followed by post hoc tests.

In assessing alignment with our message (Do you think installing nest boxes will reverse the kestrel decline?), we considered a "Yes" response as evidence that respondent disagreed with our message. Responses of "No," "I don't know," and "Other" expressed uncertainty and were thus were categorized together as agreement with our message.

We modeled correlations between our preferred outcomes and our communication strategies using linear and generalized linear models. For perception of learning, we used logistic regressions where the response variables were whether respondents perceived they learned (1)

or not (0). Similarly, we used logistic regressions for agreement with our messaging were respondents received a one if they agreed and a zero if they did not. We used linear regression to examine correlations between participation scores and communication strategies. For each response variable, we performed all-subsets model selection of a global model that contained all predictor variables. Predictor variables in each global model included the rankings of each form of communication (i.e., the KestrelCam, newsletters, discussion board, and social media). For perception of learning and agreement with our message we also used participation score in the global model. We used the dredge() function in the package MuMIn [30] to create all subsets of the global models then rank and compare them using Akaike's Information Criterion [31] corrected for small sample size [32]. We then model averaged over the entire model set [33] MuMIn's model.avg() function.

There are myriad recommendations for performing and interpreting results from analyses implementing multi-model inference. For example, the output from model.avg() function includes both the conditional and full model-averaged coefficient statistics. The full averages result from averaging across the full model set, whereas the conditional averages represent the average of each coefficient only from the models in which that coefficient is included [30, 33, 34]. It is unclear in which situations the full versus the conditional averages are preferable [35], but the full average is more conservative in that it returns lower effect sizes of predictor variables with weak effects [35]. The full method is thus useful to determine which predictor variables have the strongest effects, whereas if a factor of interest has a weak effect, the conditional average is preferable [35].

Researchers must also choose the confidence intervals with which to base inference when interpreting results of model selection. Covariates with 85% confidence intervals that exclude zero generally lower AIC values of the models in which they occur [33, 36]. Arnold [36] therefore argued for the use of 85% confidence intervals when interpreting the results of models selection performed using AIC. To allow for inference using both Arnold's [36] recommendation and traditionally-used 95% confidence intervals, some studies interpret those instances where 85% or 95% confidence intervals exclude zero as weak and strong evidence, respectively [37–39].

Given the relative paucity of studies examining outreach strategies of citizen science programs, we want to ensure that inference from our study provides guidance for other programs while also revealing potentially fruitful lines of future research. We therefore interpret our results on a continuum from strong evidence in which 95% confidence intervals of full averages exclude zero to moderate evidence where 95% confidence intervals of the conditional average excludes zero, to weak evidence where 85% confidence intervals of conditional averages exclude zero.

We used linear regressions to examine whether participation scores was correlated with perception of learning, agreement with our message, watching the KestrelCam, and following on social media.

## Results

A total of 542 respondents completed the survey for a 14.6% response rate. There was a 21.3% response rate (n = 262 respondents) among AKP partners, 9.9% response rate (n = 230 respondents) from people who signed up through The Peregrine Fund's website to get AKP or KestrelCam newsletters, and a 33.3% response rate (n = 50 respondents) from recipients that were both.

### Demographics

Overall, 50.9% of respondents identified as male, 47.8% as female (the remainder chose 'other' or chose not to answer). Regarding age, 4.8% were under 30 years old, 9.6% between 31 and

42, 18.6% between 43 and 54, 31.1% between 55 and 64, and 34.1% 65 and over (the remainder chose not to answer). Regarding highest attained education level, 4.9% had completed high school, 16.2% had completed some college, 22.0% had completed an undergraduate degree, and 55.0% had completed a graduate degree as their highest education level attained (the remainder chose not to answer).

## Communication preference

Of 542 respondents, 48.7%, 21.1%, 7.5%, 5.7%, 7.0%, and 9.7% had participation scores of 0 through 5, respectively (median = 1). Regarding communication preferences, there were differences in ranks by communication type (Kruskal-Wallis chi-squared = 259.19, df = 3, p < 0.0001). Respondents ranked newsletters (n = 529, mean ± sd = 3.9±1.2, median = 4), KestrelCam (n = 522, mean ± sd = 3.6±1.4, median = 4), discussion board (n = 209, mean ± sd = 2.8 ±1.3, median = 3), and social media (n = 511, mean ± sd = 2.6 ± 1.5, median = 2) as most to least important, respectively (Fig 1).

There was moderate evidence of a positive correlation between the ranking of the newsletter and the perception of learning. The only outcome correlated with social media was agreement with our conservation science message, indicating strong evidence for a negative correlation. There was weak evidence of positive correlations between the discussion board and both perception of learning and participation score. The KestrelCam was positively associated with perception of learning (strong evidence), and negatively associated with participation score (moderate evidence), and agreement with our conservation science message (weak evidence). Finally, there was strong evidence for a positive correlation between the perception of learning and participation score (Fig 2, S1 Table).

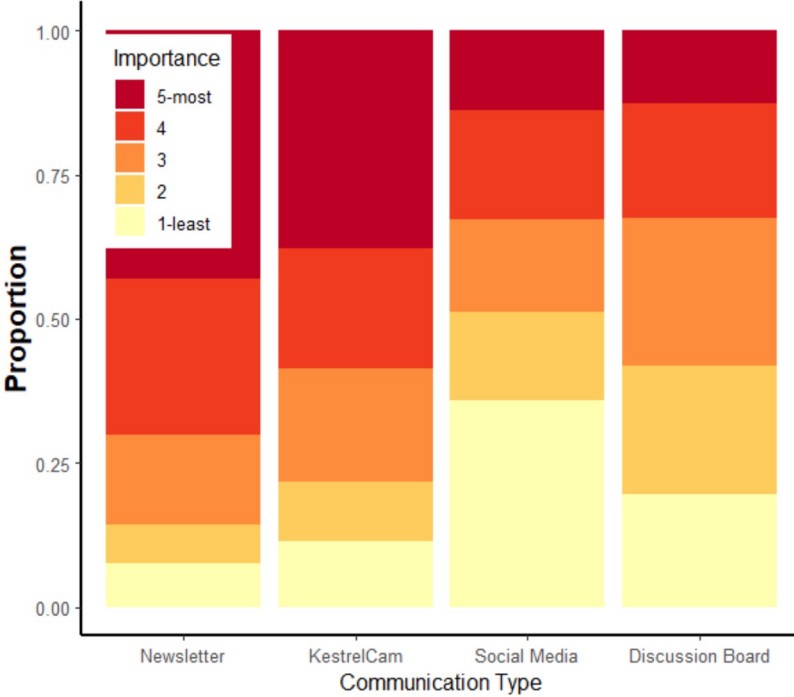

**Fig 1. Communication preferences by constituents of the American Kestrel Partnership.** 542 survey respondents ranked the level of importance of our various communication methods including newsletters, KestrelCam, social media, and our discussion board by giving each a score of between 1 (least important) and 5 (most important). Each stacked bar shows the proportion of each score value for each communication type.

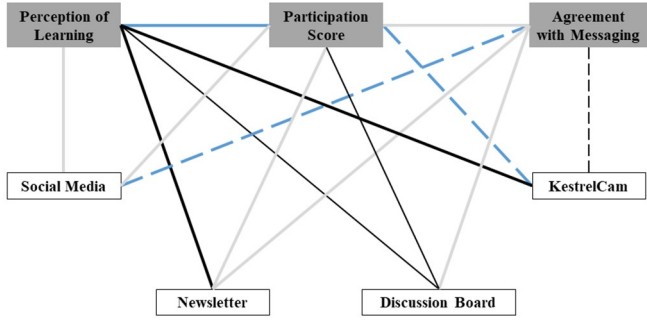

**Fig 2. Correlations between program outcomes (grey boxes) and communication types (white boxes).** Solid and dashed lines represent positive and negative correlations, respectively. Thick black lines represent correlations for which there is strong evidence, blue lines represent moderate evidence and thin black lined represent weak evidence. Grey lines represent no evidence for a correlation.

## Discussion

Our study lends insight into the efficacy of oft-used tools in outreach for citizen science efforts. Specifically, we demonstrate how social media, webcams, discussion boards, and newsletters are associated with perception of learning, agreement with our messaging, and participation in box monitoring. Our citizen science program is designed to uncover the cause of the mysterious decline of the American Kestrel and thus recruitment and retention of participants who will follow our protocol was a main goal of our outreach efforts. The perception of learning is one of our preferred outcomes only because participants often join citizen science programs out of a desire to learn [8–10] and the prospect of learning helps to retain participants [11]. Finally, because poorly placed or unmonitored nest boxes can harm populations [28, 29] or bias trends in occupancy [29], we thought it important our constituents agree with our conservation science messaging regarding the uncertainty in the efficacy of nest boxes as tools to reverse the kestrel decline.

None of our outreach methods were positively correlated with agreement with our conservation science messaging, suggesting that we should completely reevaluate our messaging strategies. However, three of our outreach methods were positively associated with perception of learning, which was in turn was positively associated with participation score. Our results therefore reveal both strengths and weaknesses of our outreach strategy while suggesting potentially fruitful avenues of future research.

Overall, respondents ranked the newsletter as their preferred communication method, and those who felt they had learned ranked the newsletter higher than those who felt they did not learn. That the newsletter ranked highest among the outreach methods might be expected because we specifically solicited our newsletter email list. Yet, given that newsletters are commonly used by citizen science programs [40–42], it is encouraging we found preference for newsletters is correlated with the perception of having learned.

We also sent the survey to people who had signed up for email updates regarding the KestrelCam, the second-highest ranked communication method. The ranking of the KestrelCam was correlated with all of our preferred outcomes, but was negatively correlated with agreement with our conservation science message and, importantly, with participation scores. Preference for the KestrelCam was positively correlated with perception of learning. It might seem counterintuitive that preference for the KestrelCam is negatively correlated with participation score yet positively associated with perception of learning, which is in turn positively correlated with participation score [43]. This seeming contradiction [43] is likely due to KestrelCam

ranking being partially correlated with perception of learning while also being driven by other factors that conversely affect participation score and less-so perception of learning. Alternatively, two mostly separate groups of respondents indicated that they either participated in box monitoring or ranked the KestrelCam high and both of those groups indicated they perceived that they learned, while a third group indicated they did not participate in box monitoring, did not value the KestrelCam, and also did not feel they learned.

Of great concern are the negative associations of the importance of the KestrelCam with participation score and agreement with our conservation science messaging. These results suggest that we should reconsider the costs and benefits of maintaining the KestrelCam. However, these correlations are supported by moderate-to-weak evidence from our analysis. Further research on webcams is therefore needed [17], and individual programs should evaluate their own goals and costs when embarking on webcams.

Online discussion forums are often hosted by citizen science programs and can enhance participant learning [15, 16]. Our results support these past studies in that the ranking of our online discussion forum was positively correlated with both perception of learning and participation score. All of our educational material is available on the discussion forum, and participants often post questions and updates. This availability of information might have contributed to perception of learning by the participants. Although anyone can read material posted to the forum, one must login to post. The correlation with participation score might therefore be due to the slightly increased effort needed to participate, especially compared to social media, the newsletter and the KestrelCam.

Survey respondents ranked social media as the least important communication type. Further, there was a negative association between the importance of social media and agreement with our messaging. We therefore found no evidence that our social media efforts have contributed positively to any project goals. Social media are ubiquitous across much of humanity and thus present unparalleled opportunities for scientific education and outreach [12], and thus we have noticed personally that many citizen science programs maintain active social media presence. However, our results demonstrate there is much to learn about effective use of social media in achieving program goals. Some social media platforms provide quantitative metrics on follower engagements. Future research should endeavor to connect such metrics with program goals.

We interpret our results under the caveat that these are correlations and thus we cannot determine causation. For example, the negative relationship between social media ranking and agreement with our messaging is perhaps evidence that our social media strategy is counterproductive to our goals. However, it is also possible our social media audience is simply more likely to disagree with our conservation science messaging. Another caveat is that we interpret our results using a spectrum of strength of inference. The two correlations in which we have the most confidence are the positive associations between the KestrelCam and newsletter with perception of learning. We have moderate confidence in the negative correlations between social media and agreement with our conservation science messaging and between the KestrelCam and nest box monitoring. We suggest these negative correlations are mostly applicable to our specific program but that they should spur other programs to examine their own strategies. Finally, there was weak evidence of correlations between the importance of the discussion board with nest box monitoring and perception of learning. These correlations highlight the importance of future research examining the efficacy of discussion boards for outreach by citizen science programs.

Our results thus help us to improve our outreach methodology. For example, when implemented we intended that the KestrelCam would encourage participation in the box monitoring program. The negative association between having a box and watching the KestrelCam provides evidence that the KestrelCam was not meeting an important objective

(recruitment) and that our investment in it should be reconsidered. Further, our results suggest areas where other citizen science programs might improve their outreach efforts and at minimum, should encourage other programs to evaluate their outreach efforts. Finally, our results highlight future research priorities. It is important for citizen science programs to periodically evaluate their communication strategies to ensure they are achieving their goals [23]. Communication tools and methods including discussion boards, webcams, newsletters, and social media can be used in myriad ways and thus each program should examine their own particular strategies. As citizen science continues to expand and mature as a discipline, so must the methodology used to recruit, retain, and educate participants.

## Conclusion

Our early outreach strategy was less focused, but a necessary phase of our maturation. This survey was the first attempt to evaluate how our constituency interacted with our program components. From it we learned that the KestrelCam was not meeting our recruitment objectives; this finding paired with expensive annual investments in the KestrelCam, led us to discontinue the KestrelCam. Survey results also showed us we needed to adjust our expectation of our other outreach components. For example, in early years of the program we had a more naïve expectation that partners would gain similar knowledge or inspiration by interacting with our content, regardless of the platform. Now, thanks to this survey plus continued research on outreach, our expectations, and thus our outreach strategy has matured.

We now approach our outreach components like a ladder leading our constituents higher and higher in engagement, borrowing loosely from Arnstein's [44] "Ladder of Citizen Participation" and future adaptations of the concept [45]. In our currently employed outreach strategy, lower rungs of the ladder, such as social media engagement, serve to bring awareness, spark interest, and provide the information needed for fans to take the next step up the ladder. The next step up, our website, provides myriad, in-depth text and graphics-based information on American Kestrels and provides the information needed to take the next step up the ladder. The next step up, registering a box and nest observation, facilitates partners going into the field, installing their own nest box, and making observations on free living kestrels, thus embracing their role as citizen scientists. We analyze their collective data and report back to them the information they have helped us learn about kestrels.

We share our experience in hopes that we can help others who may need to adopt citizen science methodologies. We encourage those program leads to research what has worked and not worked in other similar programs, bring citizen science and outreach experts on board early in program design, and work with program evaluation experts in improving upon program investments.

## Supporting information

**S1 Appendix. Full survey.** Text for full survey instrument used to evaluate outreach strategy for The Peregrine Fund's American Kestrel Partnership. Text of survey was entered into the online survey software platform, Qualtrics, such that a skip logic questioning design was employed. For example, a "no" answer on Q1 would bring Q12 up as the next question that respondent sees.
(PDF)

**S2 Appendix. Survey data.** Data from survey responses. Free form text answers were not analyzed and are thus not included here.
(CSV)

**S1 Table. Full and conditional model-averaged estimates from linear (Participation score) and generalized linear models (Perception of learning and agreement with message).** For each response variable (first column), we present the coefficients (β), standard errors (SE), z values (z), and p values (p) of the predictor variables.
(PDF)

## Acknowledgments

This work is a product of The Peregrine Fund's American Kestrel Partnership (AKP), found online at kestrel.peregrinefund.org. We express gratitude toward all of our partners and to respondents of this survey. Participation in the survey was contingent on respondents indicating they were over 18 years old and providing informed consent to participate in the study. We acknowledge the contributions of the students in the Masters of Public Administration Capstone class at Boise State University for input regarding survey development and interpretation. We thank the editorial team of *Citizen Science*: *Theory and Practice* for feedback on an earlier version of the manuscript.

## Author Contributions

**Conceptualization:** Sarah E. Schulwitz, Christopher J. W. McClure.

**Data curation:** Sarah E. Schulwitz, Greg C. Hill, Vanessa Fry.

**Formal analysis:** Sarah E. Schulwitz.

**Funding acquisition:** Sarah E. Schulwitz, Christopher J. W. McClure.

**Investigation:** Sarah E. Schulwitz, Greg C. Hill, Christopher J. W. McClure.

**Methodology:** Sarah E. Schulwitz, Greg C. Hill, Vanessa Fry, Christopher J. W. McClure.

**Project administration:** Sarah E. Schulwitz, Greg C. Hill.

**Software:** Greg C. Hill, Vanessa Fry.

**Supervision:** Sarah E. Schulwitz, Greg C. Hill, Christopher J. W. McClure.

**Visualization:** Christopher J. W. McClure.

**Writing – original draft:** Sarah E. Schulwitz, Christopher J. W. McClure.

**Writing – review & editing:** Sarah E. Schulwitz, Greg C. Hill, Vanessa Fry, Christopher J. W. McClure.

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
