## [Decision Letter · Decision Letter 0]

9 Feb 2021

PONE-D-20-40981

Evaluating Citizen Science Outreach: A case-study with The Peregrine Fund’s American Kestrel Partnership

PLOS ONE

Dear Dr. Schulwitz,

Thank you for submitting your manuscript to PLOS ONE. After careful consideration, we feel that it has merit but does not fully meet PLOS ONE’s publication criteria as it currently stands. Therefore, we invite you to submit a revised version of the manuscript that addresses the points raised during the review process.

We look forward to receiving your revised manuscript.

Kind regards,

Daniel de Paiva Silva, Ph.D.

Academic Editor

PLOS ONE

Additional Editor Comments:

Dear Sarah Schulwitz et al.,

Congrats!

After careful considerations from two independente reviewers, they find your manuscript is almost already after you take care of the very minor issues both of them raised. Since the change you need to do it very minor, I will give you one month to complete the changes, but I think you will be able to solve them a lot faster than that. Therefore, please resubmit you contribution with a rebuttal letter up to March 8th 2021.

Best regards and keep safe,

Daniel Silva

Reviewers' comments:

Reviewer's Responses to Questions

**Comments to the Author**

1. Is the manuscript technically sound, and do the data support the conclusions?

Reviewer #1: Yes

Reviewer #2: Yes

2. Has the statistical analysis been performed appropriately and rigorously? 

Reviewer #1: Yes

Reviewer #2: I Don't Know

3. Have the authors made all data underlying the findings in their manuscript fully available?

Reviewer #1: Yes

Reviewer #2: Yes

4. Is the manuscript presented in an intelligible fashion and written in standard English?

Reviewer #1: Yes

Reviewer #2: Yes

5. Review Comments to the Author

Reviewer #1: The paper assesses the usefulness of different communication tools in order to reach the projects goals - which are not merely scientific but comprise also environmental education.

A bit strange in the beginning is the "fundraising". While important for the NGO, that is no scientific or outreach target, and in fact later it is not referred to that target. That should be clarified.

Is is very interesting to see how the authors came to their disturbing results: If not properly managed, artificial nesting sites can even harm the kestrels.

The paper honestly show that participants have also different targets and resources to participate. I can completely understand the conclusions.

Reviewer #2: This is a really timely and intriguing paper that can provide lots of value and insight for groups hoping to mount effective outreach campaigns around citizen science efforts. To me the results were somewhat surprising, but perhaps underscore the decreasing relevance of social media as a source of reliable information for its consumers. There is a lot of time and effort going into social media these days, with groups scrambling to figure out how best to leverage that platform for their own messaging. This paper shows that more direct communication with user communities is more effective, and that's a really important outcome.

I'm not qualified to comment on the statistical rigor of the analysis, but it seems fine to me. Below are some minor points:

Line 13 Abstract—Typo ‘conducing’

Line 84—Can just use acronym AKP here

Thoughts tangential to this particular paper:

It seems somewhat confounding generally, to try to study the cause of a population decline, while simultaneously prompting people (and lots of them) to create new man-made spaces in the form of nest boxes for these birds to breed. Adding so many nest sites for this species might in itself affect population size, and the outcome of nesting in natural cavities might differ drastically from those in man-made boxes.

6. PLOS authors have the option to publish the peer review history of their article (what does this mean?). If published, this will include your full peer review and any attached files.

Reviewer #1: No

Reviewer #2: No

---

## [Author Response · Author response to Decision Letter 0]

5 Mar 2021

***Our responses to each comment or concern are indicated with an asterisk. Thank you! - Sarah

PLOS ONE Decision: Revision required [PONE-D-20-40981] - [EMID:97d335e60421fdf0]

PONE-D-20-40981

Evaluating Citizen Science Outreach: A case-study with The Peregrine Fund’s American Kestrel Partnership

PLOS ONE

Dear Dr. Schulwitz,

Thank you for submitting your manuscript to PLOS ONE. After careful consideration, we feel that it has merit but does not fully meet PLOS ONE’s publication criteria as it currently stands. Therefore, we invite you to submit a revised version of the manuscript that addresses the points raised during the review process.

We look forward to receiving your revised manuscript.

Kind regards,

Daniel de Paiva Silva, Ph.D.

Academic Editor

PLOS ONE

Additional Editor Comments:

Dear Sarah Schulwitz et al.,

Congrats!

After careful considerations from two independente reviewers, they find your manuscript is almost already after you take care of the very minor issues both of them raised. Since the change you need to do it very minor, I will give you one month to complete the changes, but I think you will be able to solve them a lot faster than that. Therefore, please resubmit you contribution with a rebuttal letter up to March 8th 2021.

Best regards and keep safe,

Daniel Silva

***Thank you. We have revised the formatting as described in these two links, including revising the formatting of in-text citations and the references section. 

Reviewers' comments:

Reviewer's Responses to Questions

Comments to the Author

1. Is the manuscript technically sound, and do the data support the conclusions?

Reviewer #1: Yes

Reviewer #2: Yes

2. Has the statistical analysis been performed appropriately and rigorously?

Reviewer #1: Yes

Reviewer #2: I Don't Know

3. Have the authors made all data underlying the findings in their manuscript fully available?

Reviewer #1: Yes

Reviewer #2: Yes

4. Is the manuscript presented in an intelligible fashion and written in standard English?

Reviewer #1: Yes

Reviewer #2: Yes

5. Review Comments to the Author

Reviewer #1: The paper assesses the usefulness of different communication tools in order to reach the projects goals - which are not merely scientific but comprise also environmental education.

A bit strange in the beginning is the "fundraising". While important for the NGO, that is no scientific or outreach target, and in fact later it is not referred to that target. That should be clarified.

***Thanks for catching this. We removed the two mentions of fundraising. Line 28 (in the abstract) and Line 39 (1st paragraph). 

Is is very interesting to see how the authors came to their disturbing results: If not properly managed, artificial nesting sites can even harm the kestrels.

The paper honestly show that participants have also different targets and resources to participate. I can completely understand the conclusions.

***Thank you for your comments. 

Reviewer #2: This is a really timely and intriguing paper that can provide lots of value and insight for groups hoping to mount effective outreach campaigns around citizen science efforts. To me the results were somewhat surprising, but perhaps underscore the decreasing relevance of social media as a source of reliable information for its consumers. There is a lot of time and effort going into social media these days, with groups scrambling to figure out how best to leverage that platform for their own messaging. This paper shows that more direct communication with user communities is more effective, and that's a really important outcome.

***Thank you for your comments. 

I'm not qualified to comment on the statistical rigor of the analysis, but it seems fine to me. Below are some minor points:

Line 13 Abstract—Typo ‘conducing’

***Good catch. We’ve corrected the typo to ‘conducting’ (now, Line 21)

Line 84—Can just use acronym AKP here

***We’ve included the acronym in all places that mentioned American Kestrel Partnership after the first mention. Line 28 in the abstract; Line 86 in the body.

Thoughts tangential to this particular paper:

It seems somewhat confounding generally, to try to study the cause of a population decline, while simultaneously prompting people (and lots of them) to create new man-made spaces in the form of nest boxes for these birds to breed. Adding so many nest sites for this species might in itself affect population size, and the outcome of nesting in natural cavities might differ drastically from those in man-made boxes.

***We agree with the reviewer and we don’t actually directly encourage people to put up boxes. The AKP’s nuanced perspective is that if people want to help kestrels and they want to put up a box, we serve as a resource for them to obtain the best practices of nest installation, maintenance and monitoring, and data stewardship. It is an interesting position to be in. 

6. PLOS authors have the option to publish the peer review history of their article (what does this mean?). If published, this will include your full peer review and any attached files.

Do you want your identity to be public for this peer review? For information about this choice, including consent withdrawal, please see our Privacy Policy.

Reviewer #1: No

Reviewer #2: No

PONE-D-20-40981R1

Evaluating Citizen Science Outreach: A case-study with The Peregrine Fund’s American Kestrel Partnership

Dr Sarah Schulwitz

Dear Dr Schulwitz,

Thank you for submitting your manuscript entitled "Evaluating Citizen Science Outreach: A case-study with The Peregrine Fund’s American Kestrel Partnership" to PLOS ONE. Your manuscript files have been checked in-house but before we can proceed we need you to address the following issues:

1) Thank you for including your ethics statement on the online submission form: 

"The study was exempt from IRB review due to it being research that included a survey in which responses were obtained in a manner in which the identity of the human subjects was not able to be ascertained directly or through identifiers linked to the respondents. In addition, any disclosure of the responses would not reasonably place the respondents at risk. ". 

To help ensure that the wording of your manuscript is suitable for publication, would you please also add this statement at the beginning of the Methods section of your manuscript file.

Your manuscript has been returned to your account. Please log on to PLOS Editorial Manager at https://www.editorialmanager.com/pone/ to access your manuscript.

Your manuscript can be found in the "Revisions Sent Back to the Author" link under the New Submissions menu. After you have made the changes requested above, please be sure to view and approve the revised PDF after rebuilding the PDF to complete the resubmission process.

Please note that these changes have been requested to comply with submission guidelines and your manuscript will *not* be sent to review until you have fully adhered to our requests. Once your paper has been seen by an Editor we may return it to you for further information or amendments.

We ask that you address this request within 21 days. If you require additional time, please email the journal office. We are happy to grant extensions of up to one month past this due date. If we have not heard from you within 21 days, your manuscript will be withdrawn from Editorial Manager.

Kind regards,

Anna Fodor

PLOS ONE

***We have added the statement requested at the beginning of the materials section.

---

## [Editor Report · Decision Letter 1]

9 Mar 2021

Evaluating Citizen Science Outreach: A case-study with The Peregrine Fund’s American Kestrel Partnership

PONE-D-20-40981R1

Dear Dr. Schulwitz,

We’re pleased to inform you that your manuscript has been judged scientifically suitable for publication and will be formally accepted for publication once it meets all outstanding technical requirements.

Kind regards,

Daniel de Paiva Silva, Ph.D.

Academic Editor

PLOS ONE

Additional Editor Comments (optional):

Dear Schulwitz et al.,

I am please to inform your manuscript is formally accepted for publication in PLoS One! Congratulations! PLoS One's staff will soon contact you regarding the publication guidelines.

All the best and stay safe!

Daniel Silva
---

## [Editor Report · Acceptance letter]

19 Mar 2021

PONE-D-20-40981R1 

Evaluating citizen science outreach: A case-study with The Peregrine Fund’s American Kestrel Partnership 

Dear Dr. Schulwitz:

I'm pleased to inform you that your manuscript has been deemed suitable for publication in PLOS ONE. Congratulations! Your manuscript is now with our production department. 

Kind regards, 

on behalf of

Dr. Daniel de Paiva Silva 

Academic Editor

PLOS ONE